# COVID-19 and Prostatitis: A Review of Current Evidence

**DOI:** 10.3390/diseases12070157

**Published:** 2024-07-15

**Authors:** Datesh Daneshwar, Yemin Lee, Abid Nordin

**Affiliations:** 1Urology Clinic, Prince Court Medical Centre, 39, Jalan Kia Peng, Kuala Lumpur 50450, Malaysia; 2MedCentral Consulting, International Youth Centre, Jalan Yaacob Latiff, Bandar Tun Razak, Cheras, Kuala Lumpur 56000, Malaysia; chlorinelee91@gmail.com; 3Graduate School of Medicine, KPJ Healthcare University, Nilai 71800, Negeri Sembilan, Malaysia; m.abid@kpju.edu.my

**Keywords:** COVID-19, SARS-CoV-2, ACE2, TMPRSS2, prostatitis

## Abstract

Coronavirus disease 2019 (COVID-19), a highly contagious viral disease caused by severe acute respiratory syndrome coronavirus 2 (SARS-CoV-2), poses a global health threat. The virus enters host cells by binding with angiotensin-converting enzyme 2 (ACE2), which is then facilitated by the protease activity of transmembrane serine protease 2 (TMPRSS2). It triggers a cytokine storm that eventually leads to cell apoptosis, tissue damage, and organ failure. Therefore, any organs in the human body that have both receptors are highly susceptible to COVID-19 infection, potentially resulting in multiple-organ failure. The prostate has been reported to express high levels of ACE2 and TMPRSS2. While there are limited studies regarding the association between COVID-19 and prostatitis, the possibility that SARS-CoV-2 could cause prostatitis cannot be denied. Thus, through this review, a better insight into the associations of SAR-CoV-2 can be provided.

## 1. Introduction

Coronavirus disease 2019 (COVID-19) is a highly contagious viral disease first reported in Wuhan, Hubei province, China, in December 2019. On 11 March 2020, the World Health Organization declared the outbreak of COVID-19 as a global pandemic, reporting community-scale transmission occurring in every continent on Earth [1]. As of December 2022, the pandemic had caused 6,656,601 deaths around the world, 651,918,402 individuals worldwide had been infected with COVID-19 [2]. 

Prostatitis is a medical condition associated with a frequent painful situation involving inflammation in the prostate gland and its surrounding areas. The National Institutes of Health (NIH) Classification of Prostatitis categorizes prostatitis into four main groups: acute bacterial prostatitis (Category I), chronic bacterial prostatitis (Category II), chronic prostatitis/chronic pelvic pain syndrome (Category IIIA and IIIB), and asymptomatic prostatitis (Category IV) [3]. This classification system has been widely accepted and utilized in clinical practice and research [4]. The NIH Chronic Prostatitis Symptom Index is a key tool used to evaluate symptomatology in individuals with prostatitis [5]. Prostatitis is the most common urinary tract disorder in men younger than 50 and the third most common for men older than 50 years of age [6]. According to an epidemiology study on prostatitis involving 10,617 men, the average occurrence rate of prostatitis was 8.2%, with a prevalence ranging from 2.2% to 9.7% worldwide. In the US, about 9–16% of the male population claimed to have had prostatitis-like symptoms. In Europe, about 14.2% of men had had prostatitis. Meanwhile, in Asia, the prevalence rate was 8.7% in Malaysia and 2.7% in Singapore [7]. 

Since prostatitis often causes pain in the groin and pelvic area, particularly during urination, men affected by the disease tend to experience significant impairment in their health-related quality of life (QOL) [8]. Prostatitis encompasses a spectrum of conditions, including acute and chronic bacterial prostatitis, chronic pelvic pain syndrome, and asymptomatic inflammation [9]. Acute bacterial prostatitis presents with sudden-onset symptoms, such as fever, chills, dysuria, and urinary frequency, often requiring immediate medical attention [10]. On the other hand, chronic bacterial prostatitis and chronic pelvic pain syndrome are characterized by recurrent urinary tract infections, pelvic or perineal pain, and lower urinary tract symptoms, with chronic prostatitis/chronic pelvic pain syndrome being the most prevalent form [11]. The distinction between acute and chronic prostatitis lies in the duration and persistence of the symptoms, with acute cases being more severe and sudden, while chronic cases are recurrent and long-lasting [9]. Additionally, chronic prostatitis can lead to lower urinary tract symptoms due to smooth muscle contraction in the prostate and bladder neck, along with chronic pelvic pain [11]. Therefore, research on this disease should not be overlooked. 

Severe acute respiratory syndrome coronavirus 2 (SARS-CoV-2) is the causative agent of COVID-19 [12]. It is an enveloped viral particle with a diameter ranging from 60 to 100 nm and a genome size of 29.9 kb [13]. The virus contains a nucleoprotein (N) wrapping single-stranded sense ribonucleic acid (RNA), forming a coiled tubular structure called a nucleocapsid. Associated with the nucleocapsid are several structural proteins, including the spike glycoprotein (S), the membrane (M) protein, and the envelope (E) protein. The virus genome has 5′ and 3′ terminal sequences containing five essential genes encoding for four structural proteins, with a gene order 5′-replicase open reading frame (ORF) 1ab-S-envelope (E)-membrane (M)-N-3′ [12,13]. SARS-CoV-2 enters host cells by binding its spike glycoprotein (S) to the angiotensin-converting enzyme 2 (ACE2) receptor on the host cells, facilitated by the protease activity of transmembrane serine protease 2 (TMPRSS2) [14,15]. Organs expressing both ACE2 and TMPRSS2 receptors, such as epithelial airway cells, type II pneumocytes, bronchial transient secretory cells in the lungs, and absorptive enterocytes in the small intestine, are highly susceptible to COVID-19 infection [16]. Once inside the host cell, SARS-CoV-2 triggers a vigorous inflammatory response as part of the natural immune defense. This inflammatory response can lead to a cytokine storm, characterized by the upregulation of various pro-inflammatory cytokines. In the microenvironment of intense inflammation caused by the action of pro-inflammatory cytokines and uncontrolled viral replication, cells may undergo apoptosis, leading to tissue damage and organ failure [17]. Severe COVID-19 patients who succumb to the disease often experience multiple-organ failure involving the lungs, the small intestine, the kidneys, and other organs [18].

Naturally, the prostate, which has been widely known to co-express both ACE2 and TMPRSS2, has become the target organ for COVID-19 infection [19,20]. A study reported that club cells and hillock cells are the two highest proportion of prostate epithelial cells to co-express ACE2 and TMPRSS2 [20], while another study discovered that the prostate is an organ with the second highest expression of TMPRSS2 [19]. Moreover, studies have also shown a significant connection between COVID-19 and several proliferative diseases involving the prostate, including prostate cancer and benign prostatic hyperplasia (BPH) [19,21], both of which are associated with inflammation and abnormal tissue proliferation in the prostate. Hence, the possibility that COVID-19 might also cause prostatitis cannot be denied. Since research focusing on the association between COVID-19 infection and prostatitis is still preliminary, the exact relationship between the two still needs to be discovered. Therefore, this review aims to gather all the relevant information regarding the association between COVID-19 infection and prostatitis. This is important for future research on therapeutic strategies to preserve prostate health in men, especially during the post-COVID-19 period.

## 2. Role of the Angiotensin-Converting Enzyme 2 (ACE2) Receptor in Viral Entry and Inflammation

ACE2 is a transmembrane protein and ectoenzyme with a C-terminal domain homologous to the renal protein collectrin, facilitating amino acid transport regulation [22,23]. SARS-CoV-2 exploits ACE2 as a cell surface receptor by binding its spike glycoprotein to the N-terminal region of ACE2 (Figure 1) post proteolytic modification by host cell proteases [24,25]. This interaction enhances hydrophobic interactions and salt bridge formation, boosting the virus’s affinity for ACE2, thereby facilitating viral infectivity and propagation [26]. 

In addition, ACE2 is involved in the modulation of the renin-angiotensin system (RAS), which plays an essential role in the regulation of systemic organ inflammation. Balance in RAS is important for organ protection. Under normal conditions, the expression levels of ACE2, angiotensin 1-7 (Ang1-7), and Mas are upregulated to reduce inflammation and prevent damage to the organs [27]. In SARS-CoV-2 infection, the expression levels of ACE2, Ang1-7, and Mas are downregulated. The downregulation of ACE2, Ang1-7, and Mas leads to the upregulation of ACE, Ang II, and AT1R, which have pro-inflammatory effects on various organs [22]. Studies have shown a significant downregulation of ACE2 expression and elevated circulating Ang II levels in patients with COVID-19 compared to typical healthy individuals [28]. The levels of circulating Ang II have been found to be linearly correlated with SARS-CoV-2 load, confirming the association between COVID-19 infection and the downregulation of ACE2 [29]. The imbalance in RAS triggers chronic inflammation in the cells, which may result in the apoptosis of cells and eventually lead to organ failure. A study reported that increased mortality and worsened phenotypes in elderly patients with COVID-19 were attributed to organ failure caused by the age-related downregulation of ACE2 expression in the lungs [30]. It is not just the chronic inflammation triggered by RAS imbalance, but also the clinical symptoms of patients with COVID-19 may also be aggravated by the onset of the cytokine storm that is naturally activated by the immune system due to SARS-CoV-2 invasion and multiplication [31]. During the cytokine storm, an imbalance in Th17/Treg cell function and the overactivation of immune cells trigger the secretion of a large number of pro-inflammatory cytokines, which contribute to tissue and systemic inflammation. These deleterious inflammatory responses accelerate organ injury as patients with COVID-19 die of multiple-organ failure.

## 3. Role of the Transmembrane Serine Protease 2 (TMPRSS2) Receptor in Viral Entry and Inflammation

TMPRSS2, a type II transmembrane protein ranging from 58 to 70 kDa in size, comprises three contiguous extracellular domains—a juxtamembrane LDL receptor A (LDLRA) domain, an intermediary scavenger receptor cysteine-rich (SRCR) domain, and a carboxy-terminal serine protease (SP) domain—along with an intracellular NH2 terminus (Figure 2) [14]. This tripartite LDLRA-SRCR-SP configuration indicates that TMPRSS2 is involved in transmembrane signaling functions, including the proteolytic activation of extracellular matrix proteins and growth factors, as well as defense mechanisms against external pathogens [32]. TMPRSS2 is androgen-dependent, and, upon androgen stimulation, its 32 to 42 kDa COOH SP domain is released through an autocatalytic cleavage mechanism [33,34].

TMPRSS2 plays a crucial role as a facilitator for SARS-CoV-2 infection by utilizing its SP domain for proteolytic priming of the virus. This priming process involves cleaving the S protein of SARS-CoV-2 at the S1/S2 and S2’ sites, resulting in the division of the S protein into S1 and S2 units (Figure 3). The S1 unit binds the virus to host cells via ACE2, while the S2 unit enables the fusion of viral and cellular membranes. By acting as a proteolytic primer, TMPRSS2 enhances the virus’s affinity for ACE2, leading to a more efficient cellular attachment and a potentially more robust infection within the host cells [15,35]. Moreover, COVID-19 infection increases the expression of TMPRSS2 on the host cell’s surface. Due to the high homology between SARS-CoV-2 and SARS-CoV, the causative agent of the severe acute respiratory syndrome (SARS) pandemic which occurred in 2003 [35], a study used SARS-CoV as a reference to investigate the changes in TMPRSS2 expression in cells or animals infected with SARS-CoV-2. The results showed that the expression of TMPRSS2 slightly increased in mice lung tissues infected with SARS-CoV compared to the control mice lung tissues not infected with the virus. So, it was inferred that TMPRSS2 expression might increase after SARS-CoV-2 infection [36]. Increased TMPRSS2 expression after COVID-19 infection encourages more viral entry and increases its infectivity.

TMPRSS2 is also associated with immune infiltration related to inflammatory reactions. A study investigating changes in the expression levels of TMPRSS2 mRNA in the inflammatory tumor microenvironment noted that TMPRSS2 was involved in multiple immune-related pathways such as endocytosis, transcriptional misregulation in cancer, the p53 signaling pathway, apoptosis, and the Hippo signaling pathway. Through the correlation analysis between TMPRSS2 mRNA expression and gene markers of immune cell subtypes in tumors, it was found that the TMPRSS2 mRNA expression level was significantly correlated with the infiltrating levels of CD8+ T-cells (r = −0.345, *p* < 0.0001), CD4+ T-cells (r = −0.16, *p* < 0.0001), and macrophages (r = 0.178, *p* < 0.0001). After correcting tumor purity, the TMPRSS2 mRNA expression level was found to significantly correlate with B-cells, CD8+ T-cells, neutrophils, macrophages, Th1 cells, and Treg in the tumor. A further investigation discovered that TMPRSS2 expression was also associated with tumor-infiltrating lymphocytes (TIL), which included activated eosinophils, macrophages, natural killer T-cells, myeloid-derived suppressor cells, memory B-cells, active B-cells, regulatory T-cells, T follicular helper cells, type 1 T helper cells, type 2 T helper cells, central memory CD4 T-cells, and effector memory CD8 T-cells. These findings strongly confirmed a correlation between TMPRSS2 and immune infiltration in tumors, thus proving its role in initiating immune cell infiltration in an inflammatory setting [36]. 

In addition to the immune infiltration triggered by TMPRSS2, SARS-CoV-2 entry into the host cells triggers a series of inflammatory responses to counter virus infection. These inflammatory responses can potentiate a cytokine storm that involves the upregulation of various pro-inflammatory cytokines, including tumor necrosis factor-α (TNF-α), interleukin 1β (IL-1β), IL-2, IL-6, IL-7, IL-10, monocyte chemoattractant protein-1 (MCP-1), macrophage inflammatory protein-1α (MIP-1α), CXCL10, CCL2, and CCL3. The deleterious inflammatory responses triggered by the cytokine storm combined with uncontrolled viral replication cause apoptosis in the infected cells, leading to tissue damage and, eventually, organ dysfunction as the condition persists [17]. This condition is observable in hospitalized patients with severe COVID-19 symptoms. The plasma concentrations of TNF-α, IL-2, IL-6, IL-7, IL-10, CXCL10, CCL2, and CCL3 have been shown to be significant among patients with severe COVID-19 admitted to the ICU who show signs of organ failure compared to patients with mild symptoms not admitted to the ICU [18]. This demonstrates the deleterious effects on organ function brought about by the cytokine storm and vigorous viral replication.

## 4. The Prostate as a Target Organ of COVID-19 Infection

Given the roles of ACE2 and TMPRSS2 in SARS-CoV-2 infectivity, organs that express ACE2 and TMPRSS2 have thus become the target organs for COVID-19 infection. For example, the type II alveolar (AT2) cells of the lungs are the predominant site for ACE2 expression, and, as a result, SARS-CoV-2 infection produces the greatest damage in these cells during the course of the disease [37]. Furthermore, the elevated expression of ACE2 on vascular endothelial cells has been shown to be the cause of abnormal coagulation with a prothrombotic tendency and associated hypotension observed in patients with COVID-19 during the later stages of the disease [17]. 

Meanwhile, human epithelial airway cells in the lungs co-express both ACE2 and TMPRSS2. This causes the lungs to become the main target organ for SARS-CoV-2 infection, which results in acute lung injury and is manifested as acute respiratory disease syndrome (ARDS) in the patients [38]. In addition to the epithelial airway cells, the co-expression of ACE2 and TMPRSS2 has also been identified in type II pneumocytes, bronchial transient secretory cells, absorptive enterocytes in the small intestine, and nasal goblet secretory cells, all of which are typical sites of COVID-19 symptoms [16]. So, essentially, other organs that co-express ACE2 and TMPRSS2 like the prostate are also affected by COVID-19 infection.

In a study involving the analysis of 24,519 epithelial cells from typical human prostates using a single-cell RNA sequencing dataset available publicly, 30 cells (0.11% of epithelial cells; 0.10% of all cells) in the prostate were identified to co-express both ACE2 and TMPRSS2. These cells included 0.61% (8 out of 1312 cells) of club cells, 0.40% (10 out of 2530 cells) of hillock cells, 0.18% (4 out of 2238 cells) of luminal epithelial cells, 0.09% (2 out of 2113 cells) of stromal cells, 0.06% (1 out of 1586 cells) of endothelial cells, and 0.03% (5 out of 18,439 cells) of basal epithelial cells. Among these cells, club cells and hillock cells were the two highest proportions of prostate epithelial cells co-expressing ACE2 and TMPRSS2 [20]. As reported by another study investigating the expression profiles of TMPRSS2 mRNA in humans, the prostate has been shown to be one of the organs with the highest expression of TMPRSS2. Data from the RNA sequencing of fifty-five types of tissues and six types of blood cells obtained from HPA, GTEx, and FANTOM5 showed that the RPKM values for TMPRSS2 expression were the highest in the small intestine (75.6), followed by the prostate (68.2), the pancreas (64.5), the salivary gland (52.3), the colon (38.7), the stomach (36.7), and the lungs (20.7) [19]. As ACE2- and TMPRSS2-expressing cell lines are highly susceptible to COVID-19 infection, the prostate could be an organ potentially susceptible to SARS-CoV-2 infection [39,40]. In particular, this is the case in situations where the regulation of highly expressed TMPRSS2 is controlled by androgen. With the prostate being an environment under constant androgen stimulation, its expression of TMPRSS2 may further increase and encourage viral entry by SARS-CoV-2. As a result, the prostate may serve as a reservoir for SARS-CoV-2 infiltration and a site of inflammation [20]. 

Several studies demonstrated that COVID-19 infection worsened the progression of patients diagnosed with prostate-related proliferative diseases like prostate adenocarcinoma (PRAD) [19,41]. Patients with COVID-19 with PRAD were observed to have a higher expression of TMPRSS2 compared to those without PRAD. It was discovered through genomic analyses that PRAD tissues exhibited high expression levels of the TMPRSS2 isoforms—TMPRSS2-001 and TMPRSS2-201. These two TMPRSS2 isoforms contain both SRCR-2 and trypsin domains, which play an important role in tumorigenesis and in the SARS-CoV-2 viral entry process. Therefore, the increased expression of TMPRSS2 in PRAD tissues increases the patients’ susceptibility to COVID-19 infection, which aggravates PRAD disease progression and severely lowers patients’ survival rate [19]. 

Apart from PRAD, COVID-19 infection also reportedly affects disease progression in BPH. A prospective cohort study involving 91 patients with COVID-19 diagnosed with BPH reported that the prostate-specific antigen (PSA) level of the patients during their COVID-19 infection period was significantly higher compared to the pre-COVID-19 (*p* < 0.001) and post-COVID-19 periods (*p* < 0.001). The PSA is a glycoprotein molecule synthesized by the prostate’s epithelial cells [42]. An elevated level of PSA is often associated with either a malignant or benign condition in the prostate, where there is inflammation or damage in the tissues, such as a loss of basal cells, the impaired integrity of the basement membrane, and a disrupted luminal structure. Therefore, the elevated PSA levels in the prostate of patients with BPH during their COVID-19 infection period confirmed the deleterious effect of SARS-CoV-2 on these tissues through pro-inflammatory pathways triggered by the suppression of anti-inflammatory ACE2 and the upregulation of Ang II [21]. 

Although both the studies above are not related to prostatitis, the inflammatory effect that COVID-19 has on the prostate is clearly implied in both studies. Hence, the possibility that COVID-19 might also cause prostatitis remains high.

## 5. Studies Supporting an Association between COVID-19 and Prostatitis

The idea of a possible association between COVID-19 and prostatitis was indicated in a Letter to the Editors published by Cardona Maya and Carvajal in 2020. This work pointed out that, since ACE2 and TMPRSS2 receptors are expressed on the prostate, SARS-CoV-2 binding to these cells in the prostate can potentially cause tissue alteration and inflammation that may lead to prostatitis. The authors also claimed that there was a significant group of patients with COVID-19 who were susceptible to prostatitis. However, the number of patients involved in the study and other details were not specified in the publication [43]. 

Apart from this Letter to the Editors, numerous research articles also support the notion that COVID-19 infection may increase the risk of prostatitis in the male population. For example, a prospective cohort study involving 91 patients with COVID-19 reported a considerably higher serum PSA level during the active COVID-19 infection phase (4.34 ± 3.78 ng/mL) compared to the pre-COVID-19 infection phase (1.58 ± 1.09 ng/mL; *p* < 0.001) and the post-COVID-19 infection phase (2.09 ± 2.70 ng/mL; *p* < 0.001). With PSA being a prostate-specific protein, changes in its serum level are associated with a deterioration in the prostate’s structure, such as a loss of basal cells in the prostate tissue, the impairment of the basement membrane’s integrity, and the destruction of the normal luminal structure. An increased level of PSA serum indicates a disruption in the prostate’s tissue due to inflammation, such as in the case of prostatitis [42]. Therefore, an elevated level of PSA serum during active COVID-19 infection, as shown in the results, supports the hypothesis that the prostate is a potential target organ of SARS-CoV-2 invasion. Infection of the prostate by the virus damages the prostatic tissue through the pro-inflammatory pathway in cases where there is increased cytokine release, which eventually manifests as prostatitis. Hence, this finding proves a clear and direct connection between COVID-19 infection and prostatitis [21]. 

A case report examining the histopathologic elements in the 5.2 ng enucleated prostatic tissues isolated from a COVID-19 patient identified the presence of multiple coronavirus-like spiked viral particles with sizes ranging from 73.3 nm to 109 nm. Moreover, SARS-CoV-2 RNA has also been detected in a prostate biopsy through quantitative PCR, whereas the expression of spike proteins originating from the virus has also been identified through immunofluorescent and ultrastructural visualization. Residual SARS-CoV-2 viral particles were found, through electron microscopy (EM), to endure for four months, even after prostate enucleation. These findings provide evidence that SARS-CoV-2 not only enters prostatic tissues during the acute infection stage but can also persist beyond the initial infection stage, which might affect the normal function of the prostate. Given that the viral particles persisted for such an extended period, this situation may have been more indicative of chronic prostatitis rather than acute prostatitis, as chronic prostatitis is characterized by prolonged inflammation and infection, aligning with the observed persistence of SARS-CoV-2 for up to four months. Although the prostatic tissues isolated from the patient in the case report were diagnosed to be BPH, the presence of COVID-19 viral particles is still undeniable evidence associated with the presence of abnormalities in the prostatic tissues, such as in the case of prostatitis [44]. 

Meanwhile, a histopathology study discovered spermatogenesis abnormalities in the postmortem autopsy samples of the testes of three men with COVID-19. Lymphatic and macrophage infiltration as well as the presence of spiked SARS-CoV-2 viral particles were discovered in the cytoplasm of the interstitial cells of the testes in one of the cases, demonstrating hyalinization and the thickening of the seminiferous tubules’ basement membrane, which indicated inflammation in the testes due to COVID-19 virus invasion. An immunofluorescence analysis showed a significantly higher level of ACE2 receptor expression in the testis samples of patients with COVID-19 with impaired spermatogenesis compared to those with normal spermatogenesis. The patients with impaired spermatogenesis and higher ACE2 receptor expression were also observed to show other abnormalities, including Sertoli cell-only pathology, hypospermatogenesis, early maturation arrest, and sclerosis of the seminiferous tubules. On the contrary, the patients with lower levels of ACE2 receptor expression showed 95–100% typical spermatogenesis. The inversely proportional relationship between the level of spermatogenesis and the expression of the ACE2 receptor observed through the pathological examination suggests the possibility of damages caused by COVID-19 infection on the cells responsible for spermatogenesis. Although the study did not point out a direct influence of COVID-19 infection on the increased risk of prostatitis, its findings prove the involvement of SARS-CoV-2 in damage to the testes [45]. This damage might extend to other organs in the male reproductive system, including the prostate, which might give rise to medical conditions like prostatitis.

Another case report described an occurrence of emphysematous prostatitis characterized by gas formation in the prostate of a 35-year-old Hispanic male patient who tested positive for COVID-19. Clinical observations including the presence of a prostate abscess with infection spreading to the left seminal vesicle and the base of the penis, bilateral internal iliac vein thrombosis in the abscess, and the presence of inflamed necrotic tissue in the prostatic fossa were identified through the CT scan of the pelvis. The patient also experienced sepsis caused by Klebsiella pneumoniae with hepatic involvement other than prostatic involvement. Since the association of prostatitis with K. pneumoniae and liver cirrhosis had been reported in previous works in the literature [46,47], a direct link between emphysematous prostatitis and COVID-19 infection could not be established in this case. This was due to the patient’s hepatic involvement, as he presented with cirrhosis, hyperbilirubinemia, mild transaminitis, mild alcohol use, and obesity, as well as ocular involvement, presenting with endogenous K. pneumoniae endophthalmitis. Nonetheless, the report did not rule out the possibility of prostatitis caused by COVID-19 infection, as the prostatic abscess remained unchanged even after the administration of antibiotics (ceftriaxone, meropenem, cefepime, and ceftazidime) to eliminate K. pneumoniae [48]. 

Similarly, a case series involving seven patients with COVID-19 revealed the presence of mildly enlarged prostates in all the patients through ultrasound examinations of the volume of the prostate. The mean prostate volume was recorded to be 53 mL, with a range of 35–66 mL. SARS-CoV-2 RNA (Ct value = 40) was detected in the serum of two patients within the first week of admission for COVID-19 treatment (Day 5 to Day 10). Intriguingly, the serum PSA levels recorded in all the patients were within the normal range of 0.3–3.6 ng/mL, with a mean of 1.47 ng/mL, and no signs of prostatitis were shown. Despite no obvious signs of prostatitis, the authors suspected an occurrence of viral cystitis due to SARS-CoV-2 invasion in the urothelial cells, which might have caused local or systemic inflammation which could eventually spread to the prostate and result in the onset of prostatitis [49]. Supporting the notion that COVID-19 caused damage to some prostate features, another study reported the presence of microthrombi in patients’ prostates after fatal COVID-19 infections with mild focal interstitial mononuclear inflammation [50]. 

In another animal study, using the immunoPET (PET) probe to detect the presence of SARS-CoV-2 spiked proteins, PET signals were strongly detected in the prostate of rhesus macaque infected with COVID-19. The signal was observed to change from very strong in the 3 h scan to moderately strong in the 21 h scan. The signal was maintained in the prostate for the first two weeks of COVID-19 infection. In addition, a histopathology examination showed the infiltration of macrophages, lymphocytes, plasma cells, and monocytes in the prostate. When comparing these with the probe signals from other tissues such as the lungs, the penis, and the testicles, the mean SUV revealed that the relative intensity of the prostate’s signal was among the highest detected in all the tissues. These findings indicate the prostate to be an important target organ for COVID-19 infection. The infection of the prostate can subsequently lead to inflammation on-site (prostatitis), as evidenced by immune infiltrates in the tissues [51]. 

While the study carried out by Elsaqa et al. did not find any definite correlation between the presence of the COVID-19 genome in prostate biopsies and the severity of COVID-19 manifestations or the interval between infection and prostate surgery due to the small sample size, they did find that, in the patients who tested positive, there was prolonged positive testing of COVID-19 for 40 days, and the histological examination of their prostate tissues showed nonspecific dense inflammation, with abundant lymphocytes, plasma cells, neutrophils, and the occasional eosinophil, which suggested that there was continuous inflammation in the patients [52]. Table 1 presents a summary of the findings from previous studies.

## 6. Controversial Findings Regarding the Association between COVID-19 and Prostatitis

Despite the implication of COVID-19 infection in prostatitis, several studies claimed no association between the two incidences. For instance, in a clinical observation which measured the serum PSA level in 23 patients diagnosed with COVID-19, it was reported that the mean serum PSA level was 1.13 ng/mL. The study’s authors remarked that the PSA level was within the normal range and emphasized that COVID-19 infection was not associated with prostatitis. However, changes in the serum PSA level in the patients during the pre-COVID-19 infection phase and the post-COVID-19 infection phase were not measured and recorded. The values were only measured once during the active COVID-19 infection phase in each patient. As a result, a comparison of the changes in the serum PSA levels between the three different phases of COVID-19 infection was not possible with this limitation. This may give rise to a controversial conclusion, which fails to establish a connection between COVID-19 infection and prostatitis [53]. Similarly, a study involving ten patients infected with COVID-19 who had undergone prostate biopsy and radical prostatectomy (RP) reported the absence of gross structural histopathological damage to the prostate during COVID-19 infection. This suggested that COVID-19 infection did not cause any inflammatory changes in the prostate gland [54]. Thus, an association between COVID-19 infection and prostatitis could not be established.

Another study in China reported the absence of SARS-CoV-2 RNA in all 23 samples of expressed prostatic secretion (EPS) isolated from both eighteen confirmed and five suspected cases of COVID-19 infection. Regardless of the days of infection, the symptoms involving the urinary system, and the presence of reproductive system-related complications such as chronic prostatitis or benign prostatic hyperplasia (BPH), the EPS samples from all the patients did not contain any SARS-CoV-2 virus [55]. This was the case not just for patients with an active COVID-19 infection, but the SARS-CoV-2 virus was also not detected in all the EPS samples isolated from 74 patients who had recovered from COVID-19 infection [56]. The absence of affirmative evidence of SARS-CoV-2 virus expressed in the EPS samples of patients infected with COVID-19 suggested that the prostate may not be a target organ for COVID-19 infection and that prostatitis may not necessarily be caused by COVID-19. 

Nonetheless, the limitations mentioned in these studies might be the cause for the controversial findings that suggested no association between prostatitis and COVID-19 infection. For example, the EPS samples were only isolated from patients with mild and moderate COVID-19 symptoms and not severe symptoms. As such, the possible presence of SARS-CoV-2 virus in the EPS samples of patients with severe COVID-19 could not be confirmed, which might have caused the possibility of prostatitis occurrence in severe patients to be overlooked [55]. Moreover, analyses of specific indicators or markers reflecting inflammation in the prostate were not performed, along with the detection of SARS-CoV-2 virus in the EPS samples. Hence, the results on virus detection in the EPS samples could not be compared and confirmed with the expression of inflammation markers in the prostate, thus overlooking the possibility of prostatitis occurrence in patients with COVID-19 [56]. Given the limitations of these studies, it is still too early to conclude that prostatitis is not associated with COVID-19 infection. Further research covering all these limitations should be conducted to provide more information regarding the association between prostatitis and COVID-19 infection.

## 7. Concluding Remarks and Future Perspective

Both ACE2 and TMPRSS2 receptors are essential for SARS-CoV-2 invasion during COVID-19 infection. Therefore, organs in the human body that co-express ACE2 and TMPRSS2 are highly susceptible to COVID-19 infection. The prostate, one of the many organs co-expressing ACE2 and TMPRSS2, is a target organ for COVID-19 infection. Several studies have confirmed the inflammatory effects that COVID-19 exerts on the prostate, which result in the occurrence of prostatitis among male patients. However, at the same time, there are still a handful of studies that disagree regarding the establishment of an association between COVID-19 and prostatitis due to the absence of definite evidence confirming the relationship between the two. Since the investigation of this subject is still preliminary and multiple limitations exist in the studies mentioned in this review, more research is required before the association between COVID-19 infection and prostatitis can be confirmed.

## Figures and Tables

**Figure 1 diseases-12-00157-f001:**
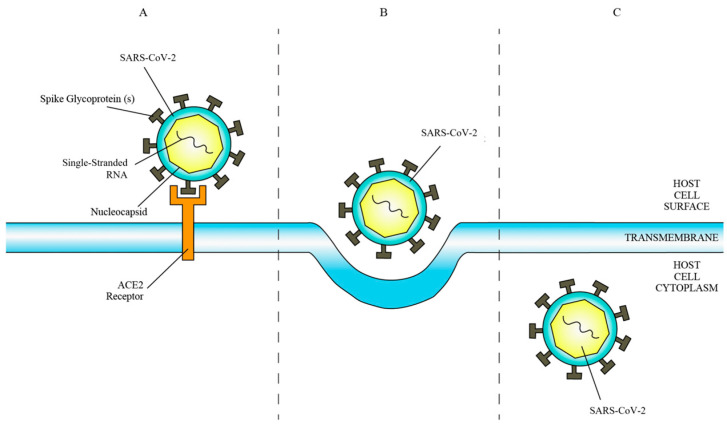
(**A**) Binding of SARS-CoV-2 to ACE2 receptor. (**B**) Fusion of SARS-CoV-2 membrane with host cell membrane. (**C**) Entry of SARS-CoV-2 into host cell cytoplasm.

**Figure 2 diseases-12-00157-f002:**
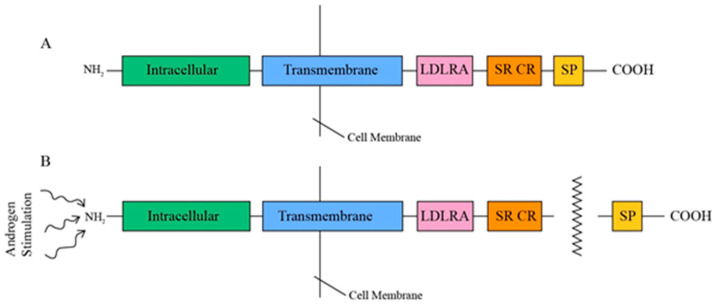
(**A**) Tripartite structure of TMPRSS2. (**B**) Release of SP proteolytic domain under androgen stimulation.

**Figure 3 diseases-12-00157-f003:**
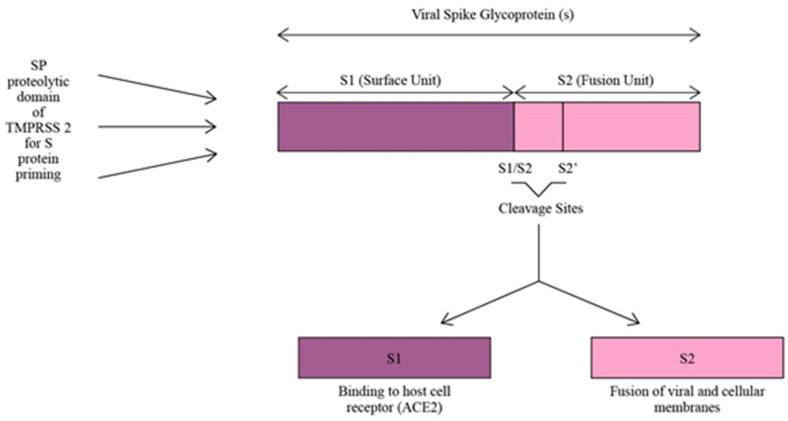
S protein priming of SARS-CoV-2 by SP proteolytic domain of TMPRSS2.

**Table 1 diseases-12-00157-t001:** Summary of the findings from previous studies.

Study	Title	Findings
Cinislioglu et al., 2022 [21]	Variation of Serum PSA Levels in COVID-19 Infected Male Patients with Benign Prostatic Hyperplasia (BPH): A Prospective Cohort Studys	Higher serum PSA levels during the active COVID-19 infection phase compared to the pre- and post-infection phases (4.34 ± 3.78 ng/mL vs. 1.58 ± 1.09 ng/mL and 2.09 ± 2.70 ng/mL, respectively). An elevated PSA indicates prostate tissue disruption and inflammation.
Cardona Maya and Carvajal, 2020 [43]	SARS-CoV-2 and prostatitis: Dangerous relationship for male sexual and reproductive health	Association between COVID-19 and prostatitis suggested due to the presence of ACE2 and TMPRSS2 receptors on the prostate.
Reddy et al., 2022 [44]	SARS-CoV-2 in the Prostate: Immunohistochemical and Ultrastructural Studies.	Presence of SARS-CoV-2 viral particles in prostate tissues and detection of SARS-CoV-2 RNA through PCR. The viral particles persisted for four months post infection, suggestive of chronic prostatitis.
Achua et al., 2021 [45]	Histopathology and Ultrastructural Findings of Fatal COVID-19 Infections on Testis.	Abnormalities in spermatogenesis observed in men with COVID-19. ACE2 receptor expression correlated with impaired spermatogenesis. No direct link to prostatitis mentioned.
Crane et al., 2021 [48]	Rare Case of Endogenous Klebsiella Endophthalmitis Associated with Emphysematous Prostatitis in a Patient with Diabetes, Cirrhosis and COVID-19.	Emphysematous prostatitis in a COVID-19 patient, complicated by sepsis and hepatic involvement. No conclusive link to COVID-19 established.
Mumm et al., 2020 [49]	Urinary Frequency as a Possibly Overlooked Symptom in COVID-19 Patients: Does SARS-CoV-2 Cause Viral Cystitis?	Mildly enlarged prostates in seven patients with COVID-19. SARS-CoV-2 RNA was detected in the serum of two patients. No signs of prostatitis but suspected viral cystitis.
Al-Nemer et al., 2020 [50]	Histopathologic and Autopsy Findings in Patients Diagnosed with Coronavirus Disease 2019 (COVID-19): What We Know So Far Based on Correlation With Clinical, Morphologic and Pathobiological Aspects.	Presence of microthrombi in the prostates of fatal COVID-19 cases.
Madden et al., 2022 [51]	An ImmunoPET Probe to SARS-CoV-2 Reveals Early Infection of the Male Genital Tract in Rhesus Macaques	PET signals detected in the prostate of COVID-19-infected macaques. Immune infiltrates observed in prostate tissues.
Elsaqa et al., 2022 [52]	Molecular detection of the COVID-19 genome in prostatic tissue of patients with previous infection	No definite correlation was found between the presence of COVID-19 genome in prostate biopsies and the severity of COVID-19 manifestations or the interval between infection and prostate surgery due to the small sample size. In patients who tested positive, prolonged positive testing for COVID-19 for 40 days. The histological examination showed nonspecific dense inflammation.

## Data Availability

No new data were created or analyzed in this study. Data sharing is not applicable to this article.

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
