# Peer review of "COVID-19 and Prostatitis: A Review of Current Evidence"

_diseases, 2024, doi:10.3390/diseases12070157_

Round 1
Reviewer 1 Report
Comments and Suggestions for Authors
Prostatitis is a multifaceted constellation of symptoms and causes. Simply addressing "prostatitis" can be misleading, since there are many forms of this condition (including forms of chronic non-bacterial pelvic pain syndrome). Unfortunately, the authors did not address the complexity of prostatitis, making it impossible to delineate the topic of their own work. The authors should define whether their work regards any form of prostatitis (including chronic and acute forms), and any cause (bacterial, viral, non-bacterial, etc.). The introduction should include the NIH classification of prostatitis, and the aim of the work should clearly state which category of prostatitis can be associated with SARS-CoV-2 infection via postulated mechanisms. The description of the role of the ACE2 and TMPRSS2 receptors is useful, but too long, containing some textbook knowledge, and not building a logical bridge to the main topic of the article: the role of SARS-CoV-2 in prostatitis via the presumed role of both receptors. The studies supporting such an association (discussed in paragraph 6) should be summarized in a table. Finally, among the limitations of the study, the narrow focus on the role of the two receptors emerges, as the role of SARS-CoV-2 and the spectrum of prostatitis (and generally LUTS) can be mediated via several other mechanisms, which remain unmentioned. The submitted manuscript, providing a juxtaposition of partly well-known information about SARS-CoV-2 and – on the other hand – partly speculative-hypothetical assumptions, requires a profound revision.
Reviewer 2 Report
Comments and Suggestions for Authors
diseases-3001874
COVID-19 and Prostatitis: A Review of Current Evidence
The manuscript summarized current evidence of the relationship between COVID-19 and prostatitis. This is a well-prepared manuscript with updated evidence. The manuscript is suitable for publication after a revision. Please consider the following comments to improve this manuscript.
1. Section 2 is too short. The authors should either combine it with section 1 or expand it further.
2. Please ensure that all figures are original. If not, please mention the source and permission to reuse.
3. Please summarize major findings into a table to help illustrate the review better.
4. Please include a section introducing prostatitis with more updated information. The current one in Section 1 is too general and brief.
Comments on the Quality of English LanguageMinor editing of English language required
Reviewer 3 Report
Comments and Suggestions for Authors
This article reviewed recently published articles related with covid19 infection and chronic prostatitis. There are several points that authors might consider to revise:
1) The authors reported the pathophysiology of covid19 infection and the mechanisms for the involvement in organ damage. There are too lengthy discussion not related to the prostatitis.
2) The reports of prostatitis ad covid19 infection are limited, and most of the reports were lack of strong and direct evidence. In reality, covid19 infection should cause acute prostatitis, not chronic prostatitis.
3) The authors should shorten the length of this review, and add more clinical evidence and data to support the association between covid19 infection and prostatitis.
Round 2
Reviewer 2 Report
Comments and Suggestions for Authors
The manuscript was appropriately revised and can be accepted.
Author Response
Comment 1: The manuscript was appropriately revised and can be accepted.
Author's Note: Thank you.
Reviewer 3 Report
Comments and Suggestions for Authors
The authors have responded adequately and the revised manuscript is comprehensive, although still redundant, but acceptable for publication in this journal.
Author Response
Comments 1: The authors have responded adequately and the revised manuscript is comprehensive, although still redundant, but acceptable for publication in this journal.
Response 1: Thank you